# An Analysis of Adherence to Vaccination Recommendations in a Thoracic Organ Transplant Cohort

**DOI:** 10.3390/vaccines8040622

**Published:** 2020-10-22

**Authors:** Deeksha Jandhyala, Jessica D. Lewis

**Affiliations:** Division of Infectious Diseases, Department of Medicine, Medical University of South Carolina, Charleston, SC 29425, USA; lewisje@musc.edu

**Keywords:** transplant, vaccine, heart transplant, lung transplant, infectious diseases, transplant infectious diseases

## Abstract

(1) Background: Vaccination of solid organ transplant (SOT) candidates and recipients is vital to decrease infection-related morbidity and mortality. Here we describe our heart and lung transplant programs’ rates of completion of hepatitis B and pneumococcal vaccinations and identify potential opportunities for improvement. (2) Methods: This is a single-center retrospective study that included all heart and lung transplant recipients between 1 July 2013 and 31 July 2018. We assessed demographics, causes of organ failure, pretransplant hepatitis B immune status, and completion rates for hepatitis B vaccine series, pneumococcal conjugate vaccine (PCV13), and pneumococcal polysaccharide vaccine (PPSV23). (3) Results: A total of 41 patients were included in the heart transplant cohort. Twelve (29.3%) had baseline hepatitis B immunity. Only 8/29 (27.6%) completed the entire 3-dose hepatitis B vaccination series pretransplant. Pretransplant PCV13 and PPSV23 vaccination rates were 58.5% (24/41) and 48.8% (20/41), respectively; no additional patients received PCV13 or PPSV23 post-transplant. In the heart transplant cohort, a majority (82.9%) of patients were evaluated by the Transplant Infectious Diseases consultative service (TxID) pretransplant, and this had a statistically significant association with increased pneumococcal vaccination rates (*p* = 0.0017, PCV13 and *p* = 0.0103, PPSV23). In total, 55 patients were included in the lung transplant cohort. Five (9.1%) had baseline hepatitis B immunity; 33/50 (66.0%) completed the hepatitis B vaccine series in the pretransplant setting. Pretransplant PCV13 and PPSV23 vaccination rate was 40.0% (22/55) and 69.1% (38/55), respectively. There was only a 47.3% and 72.3% completion rate overall in the post-transplant setting. (4) Conclusions: There continues to be a need for a comprehensive and coordinated effort to increase vaccine adherence for all SOT candidates in the pretransplant setting.

## 1. Introduction

Vaccination of solid organ transplant (SOT) candidates and recipients is vital to decreasing infection-related morbidity and mortality. Published guidelines recommend administering vaccinations prior to transplantation, as there is a diminished immunologic response to many vaccines in the post-transplant period in the setting of chronic immunosuppression [1]. While live virus vaccines are generally contraindicated post-transplant, inactivated vaccinations have been shown to be safe to administer after SOT if they are unable to be completed pretransplant [2,3]. Additionally, theoretical concerns of potentially triggering a rejection episode with vaccinations are not supported by current literature [4]. As an example, influenza vaccination post-transplantation was suggested to cause low level rejection in heart recipients, but this was not confirmed by additional studies [5].

There is a robust body of literature on vaccination adherence in adult abdominal organ transplant candidates and recipients. These reports have demonstrated that adherence to vaccination recommendations in abdominal transplant recipients is as low as 35% for the hepatitis B vaccination series [6]. Literature on vaccination adherence in thoracic organ transplant candidates and recipients is sparse.

Here, we describe rates of completion of hepatitis B and pneumococcal vaccinations in our heart and lung transplant programs and identify opportunities for improvement.

## 2. Materials and Methods

This is a single-center retrospective study performed at the Medical University of South Carolina (MUSC) (Charleston, SC, USA), a 700-bed academic tertiary care center that provides comprehensive organ transplant services. The study cohort includes all adult (>18 years old) heart and lung transplant recipients who underwent transplantation between 1 July 2013 and 31 July 2018. We assessed demographics, causes of organ failure, pretransplant hepatitis B immune status (with immunity defined as hepatitis B surface antibody (anti-HBs) titer greater than 10 mIU/mL), and completion rates for hepatitis B vaccine series, pneumococcal conjugate vaccine (PCV13), and pneumococcal polysaccharide vaccine (PPSV23). Hepatitis B vaccine series was considered to be complete if a subject received ≥3 doses of the vaccine.

Our medical center utilizes Epic© (Epic Systems Corporation, Verona, WI, USA) as its electronic medical record (EMR). Vaccination data was extracted from the EMR’s vaccination “tab”. The vaccination “tab” in the EMR is auto-populated when vaccines are administered within our medical system. Vaccination records from the state health department can be imported directly and patients’ personal records of vaccines administered outside of the medical system can be manually entered into this “tab” as well.

Routine pretransplant evaluation by the Transplant Infectious Diseases consultative service (TxID) was implemented in the heart transplant program in 2015, while lung transplant candidates continued to be evaluated by TxID on an ad hoc basis. We compared vaccination completion rates between subjects who underwent pretransplant TxID evaluation and those who did not, using Fisher’s exact test with a two-tailed *p* value < 0.05 to define statistical significance. The MUSC Institutional Review Board approved this study with a waiver of consent (IRB#00083425).

## 3. Results

A total of 41 adult heart transplant recipients and 55 adult lung transplant recipients underwent transplantation during the 61 month study period and were included in the data set. Demographics of the cohort are presented in Table 1.

In the heart transplant cohort, all 41 patients underwent pretransplant serological screening for hepatitis B immunity with anti-HBs; 12 (29.3%) had baseline immunity (Table 2a). Of the remaining 29 non-immune patients, only 8/29 (27.6%) completed the entire 3-dose hepatitis B vaccination series pretransplant; one additional patient completed the series post-transplant for a total completion rate of 31.0% (Table 2b). Only five of these subjects had post-vaccination completion anti-HBs checked: four had protective anti-HBs measured and one remained anti-HBs-nonreactive. We also assessed follow-up anti-HBs status in the 20 patients who received less than the full 3-dose series: 10 patients did not have a follow-up anti-HBs performed, eight patients were non-reactive, and two patients had protective anti-HBs measured. PCV13 and PPSV23 vaccination rates were 58.5% (24/41) and 48.8% (20/41), respectively (Table 3). In total, 39.0% of heart transplant recipients received both pneumococcal vaccines.

In the lung transplant cohort, all 55 patients underwent pretransplant serological screening for hepatitis B immunity; five (9.1%) had baseline immunity (Table 2a). Of the remaining 50 non-immune patients, 33/50 (66.0%) completed the hepatitis B vaccination series in the pretransplant setting and no additional patients completed the series post-transplant (Table 2b). In total, 20 of the 33 patients who completed the vaccine series had post-vaccination completion anti-HBs checked: 11 had protective anti-HBs measured and nine remained anti-HBs-nonreactive. We also assessed follow-up anti-HBs status in the 17 patients who received less than the full 3-dose series: 12 patients did not have a follow-up anti-HBs performed, four patients were non-reactive, and one patient had protective anti-HBs measured. PCV13 and PPSV23 vaccination rates were 47.3% (26/55) and 72.3% (40/55), respectively (Table 3). A total of 34.5% of lung transplant recipients received both pneumococcal vaccines.

Table 4 shows the association between pretransplant TxID evaluation and vaccine completion rates in the entire thoracic organ transplant cohort and in the heart and lung transplant cohorts individually. In the heart transplant cohort, 82.9% (34/41) of patients were evaluated by TxID pretransplant, and this had a statistically significant association with increased pneumococcal vaccination rates. A total of 24/34 patients (70.6%) received PCV13 if TxID evaluation was done versus 0/7 patients (0%) without TxID evaluation (*p* = 0.0017). Similarly, 20/34 patients (58.8%) received PPSV23 if TxID evaluation was done compared to 0/7 (0%) without consultation (*p* = 0.0103). No significant difference was found in hepatitis B vaccination completion rates.

In the lung transplant cohort, in which pretransplant TxID evaluation was far less frequently performed (occurring in only 12.7% of patients), no differences were seen in PPSV23 and hepatitis B vaccination rates whether or not TxID evaluation was performed. However, subjects were more likely to receive PCV13 if a TxID evaluation was not done (0% with TxID evaluation versus 45.8% without TxID evaluation, *p* = 0.0421). When the heart and lung transplant cohorts are combined, the impact of TxID evaluation is no longer statistically significant.

## 4. Discussion

SOT recipients are at increased risk of acquiring vaccine-preventable infections due to chronic immunosuppression and anatomic changes related to transplant. The importance of receiving pretransplant vaccinations is highlighted by data that show waning titers in the post-transplant period [7]. Current guidelines recommend administration of vaccinations in the pretransplant setting to optimize sero-response [1], however published data have shown that vaccination adherence, particularly in adult abdominal organ candidates and recipients, is low [8]. To our knowledge, this study is the first to assess vaccination adherence in a thoracic organ transplant cohort.

Hepatitis B vaccination has become increasingly important in the era of more frequent utilization of Public Health Service (PHS) increased risk donors [9]. There has been a more concerted effort to safely expand the donor organ pool due to the widening gap between organ availability and demand [10]. One method to expand the donor pool is to use organs from isolated hepatitis B core antibody positive (HBcAb-positive) donors. The risk of donor-derived hepatitis B infection in this scenario largely depends on the organ being transplanted (with highest risk in liver transplantation) and the sero-status of the recipient, with higher anti-HBs titers associated with lower risk of transmission of hepatitis B from isolated HBcAb-positive donors. In a post-hoc analysis of a limited-access dataset of the United Network for Organ Sharing database from 2000 to 2010, HBcAb-positive donor status did not significantly affect overall survival of thoracic transplant recipients [11]. Therefore, priority must be given to complete the hepatitis B vaccination series prior to thoracic organ transplantation to maximize efficacy and allow for the safer use of HBcAb-positive donor organs.

It is recommended that anti-HBs titers be checked approximately 4 weeks after the last dose of the hepatitis B vaccine, to ensure protective levels of anti-HBs have been achieved [1]. Just over half of our cohort had post-hepatitis B vaccination anti-HBs performed and we found significant rates of non-response, that is, a lack of development of protective immunity, as defined by an anti-HBs titer >10 mIU/mL. In the heart transplant cohort, one of five subjects (20%) who had post-vaccination anti-HBs performed was identified as a non-responder, while in the lung transplant cohort, nine of 20 subjects (45%) who had post-vaccination anti-HBs performed were identified as non-responders. Guidelines recommend revaccinating hepatitis B vaccine non-responders with either a single booster dose of the vaccine or reinitiating the entire vaccine series [1].

SOT recipients are at increased risk of invasive pneumococcal disease and thus pneumococcal vaccination in this population is important as well. Currently there are two formulations of pneumococcal vaccines that are recommended in SOT candidates and recipients, PPSV23 and the PCV13. In the post-transplant setting, kidney transplant recipients have shown a suboptimal vaccine response to pneumococcal vaccinations [12], and thus these vaccinations are recommended for administration to all SOT candidates pretransplant if at all possible.

In our review, the majority of heart and lung transplant candidates (70.7% and 90.9%, respectively) were non-immune to hepatitis B pretransplant; only 27.6% of hepatitis B non-immune heart transplant candidates completed the vaccine series pretransplant, while 66.0% of hepatitis B non-immune lung transplant candidates did. Pretransplant TxID evaluation did not impact completion rates in either cohort. Pneumococcal vaccination rates, 52.1% for PCV13 and 62.5% for PPSV23 in the entire cohort, are fairly robust, perhaps due to the single dose schedule of the vaccines, but there is clearly room for improvement. In our heart transplant population, we did see a significant improvement in pneumococcal vaccination rates when TxID evaluation was performed pretransplant.

Our study has a few limitations, primarily a reliance on documentation of vaccinations in the EMR, which often requires importation of records from outside facilities in order to be accurate and complete, and thus, vaccination rates may be underestimated. We did not review clinician notes in order to obtain additional information about vaccination administration. This is a limitation in many vaccination studies, as vaccination records are not kept in a centralized database that can be accessed from all institutions and EMRs. Routine pretransplant TxID evaluation was implemented midway through the study period in the heart transplant program, but not the lung transplant program, so the cohorts were analyzed separately as well as in a combined cohort in order to account for this disparity. In addition, as this was a single center study, results cannot necessarily be generalized to other transplant centers.

Our study is the first to our knowledge to demonstrate that vaccination adherence is suboptimal in the thoracic organ transplant population, as has been demonstrated by others in abdominal transplant recipients. Pineda et al. recently described pretransplant hepatitis B vaccination completion rates and/or documented anti-HBs sero-positivity in only 35% of liver transplant candidates, and completion rates of 57% for PCV13 and 62% for PPSV23 [6].

Creative interventions to improve vaccination adherence in this vulnerable population are needed. Some institutions have implemented EMR reminders to prompt providers to administer vaccines. For example, a pediatric kidney transplant group initiated an age-based vaccine algorithm, obtaining vaccine records, and generating reminders for patients and clinicians for pneumococcal vaccinations, resulting in significant improvement in adherence [12]. Use of the double-dose accelerated combined hepatitis A and B vaccination schedule could be used to shorten the duration necessary to complete the entire vaccine series, which may improve completion rates [13]. Other “low tech” interventions could include cards for SOT candidates to take to each transplant and primary care clinic appointment to serve as a reminder for specific vaccinations to be administered. Increased involvement of TxID in the pretransplant evaluation may be helpful, as would improvement in documentation of vaccinations given at facilities outside of the transplant center.

## 5. Conclusions

In summary, immunization is a vital component of our efforts to prevent infectious complications in the susceptible SOT population, and yet, our data demonstrate that we have much room for improvement. There continues to be a need for a comprehensive and coordinated effort to increase vaccine adherence for all SOT candidates in the pretransplant setting. Multicenter prospective interventional studies to address this problem are needed.

## Figures and Tables

**Table 1 vaccines-08-00622-t001:** Demographics of cohort of 96 thoracic organ transplant recipients.

Demographics	Heart Transplant Recipients	Lung Transplant Recipients	Entire Cohort
	*n* = 41	*n* = 55	*n* = 96
Sex			
Male, n (%)	34 (82.9)	26 (47.3)	60 (62.5)
Female, n (%)	7 (17.1)	29 (52.7)	36 (37.5)
Ethnicity			
White/Caucasian, n (%)	21 (51.2)	9 (16.4)	30 (31.3)
Black/African-American, n (%)	20 (48.8)	46 (83.6)	66 (68.7)
Age, median (range)	49.6 (18.5–69.5)	59.9 (20.1–69.0)	56.2 (18.5–69.5)
Cause of end-stage disease, n (%)			
NICM	26 (63.4)		
ICM	13 (31.7)		
Congenital heart disease	2 (4.9)		
CF		7 (12.7)	
COPD		11 (20.0)	
ILD		6 (10.9)	
NSIP		3 (5.5)	
Sarcoidosis		4 (7.3)	
Comorbidities			
Obesity (BMI > 30), n (%)	17 (41.5)	15 (27.3)	32 (33.3)
Diabetes mellitus, n (%)	13 (31.7)	10 (18.2)	23 (24.0)
Hypertension (≥140/90), n (%)	35 (85.4)	42 (76.4)	77 (80.2)
Hyperlipidemia (LDL > 100 mg/dL), n (%)	15 (36.6)	21 (38.2)	36 (37.5)
LDL, mg/dL, median (range)	83 (39–192)	89 (21–235)	87.5 (21–235)

NICM = non-ischemic cardiomyopathy; ICM = ischemic cardiomyopathy; CF = cystic fibrosis; COPD = chronic obstructive pulmonary disease; ILD = interstitial lung disease; NSIP = nonspecific interstitial pneumonia; BMI = body mass index; LDL = low-density lipoprotein.

**Table 2 vaccines-08-00622-t002:** Hepatitis B immunity and vaccination status.

(a) Pretransplant Anti-HBs Status			
	Heart transplant recipients	Lung transplant recipients	Entire cohort
	*n* = 41	*n* = 55	*n* = 96
Anti-HBs positive, *n* (%)	12 (29.3)	5 (9.1)	17 (17.7)
Anti-HBs nonreactive, *n* (%)	29 (70.7)	50 (90.9)	79 (82.3)
**(b) Vaccination Status of Hepatitis B Nonimmune Subjects (Anti-HBs-Nonreactive)**			
	**Heart transplant recipients**	**Lung transplant recipients**	**Entire cohort**
	***n* = 29**	***n* = 50**	***n* = 79**
Completed hepatitis B vaccine series pretransplant, *n* (%)	8 (27.6)	33 (66.0)	41 (51.9)
Completed hepatitis B vaccines series post-transplant, *n* (%)	1 (3.4)	0 (0.0)	1 (1.3)
Total subjects who completed hepatitis B vaccine series, *n* (%)	9 (31.0)	33 (66.0)	42 (53.2)

Anti-HBs = hepatitis B surface antibody.

**Table 3 vaccines-08-00622-t003:** Pneumococcal vaccination status.

	Heart Transplant Recipients	Lung Transplant Recipients	Entire Cohort
	*n* = 41	*n* = 55	*n* = 96
Received PCV13, *n* (%)	24 (58.5)	26 (47.3)	50 (52.1)
Received PPSV23, *n* (%)	20 (48.8)	40 (72.3)	60 (62.5)
Received both PCV13 and PPSV23, *n* (%)	16 (39.0)	19 (34.5)	35 (36.5)

PCV13 = 13-valent pneumococcal conjugate vaccine; PPSV23 = 23-valent pneumococcal polysaccharide vaccine.

**Table 4 vaccines-08-00622-t004:** Impact of transplant infectious disease evaluation on vaccination rates.

	Heart Transplant Recipients			Lung Transplant Recipients			Entire Cohort		
	TxID Evaluation	No TxID Evaluation	*p*-Value	TxID Evaluation	No TxID Evaluation	*p*-Value	TxID Evaluation	No TxID Evaluation	*p*-Value
	*n* = 34	*n* = 7		*n* = 7	*n* = 48		*n* = 41	*n* = 55	
Received PCV13, *n* (%)	24 (70.6)	0 (0.0)	0.0017	0 (0.0)	22 (45.8)	0.0421	24 (58.5)	22 (40.0)	0.1111
Received PPSV23, *n* (%)	20 (58.8)	0 (0.0)	0.0103	4 (57.1)	34 (70.8)	0.7403	24 (58.5)	34 (61.8)	0.9074
Received Hepatitis B vaccine, *n* (%)	8 (34.8)	1 (16.7)	0.7507	3 (60.0)	30 (66.7)	>0.9999	11 (39.3)	31 (60.8)	0.1102

TxID = transplant infectious diseases; PCV13 = 13-valent pneumococcal conjugate vaccine; PPSV23 = 23-valent pneumococcal polysaccharide vaccine.

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
