# Peer review of "An Analysis of Adherence to Vaccination Recommendations in a Thoracic Organ Transplant Cohort"

_vaccines, 2020, doi:10.3390/vaccines8040622_

Round 1

Reviewer 1 Report

The revised manuscript has been improved. I have no further comments.

Reviewer 2 Report

Dear,  I have just reviewed the revised highlighted version and the authors' response letter. I believe this paper carries a significant message to the transplantation world that is dealing with patients under immunosuppression. I am satisfied with the revision and the response. Recommend approval for publication in your journal Vaccines.

This manuscript is a resubmission of an earlier submission. The following is a list of the peer review reports and author responses from that submission.

Round 1

Reviewer 1 Report

  1. Please include what Hep B vaccines were used and the dose.
  2. What were the titres in the immune patients? 
  3. Was there any association with occult hepatitis B in those with or without anti-HBs?
  4. Please provide detailed clinical demographics tables for your cohorts that includes ethnicity, ALT, AST, metabolic syndromes.
  5. Make sure to spell out the abbreviations when you first use them.

Author Response

Thanks for your valuable comments

  1. Please include what Hep B vaccines were used and the dose.

Could the reviewer please clarify what information they would like included? Does the reviewer want us to include general information on the type and dose of hepatitis B vaccines used at our institution in the Methods section or more specifically how many subjects received each type and dose of vaccine?

  1. What were the titres in the immune patients? 

Our clinical laboratory reports hepatitis B surface antibody (anti-HBs) results as a qualitative “positive” or “non-reactive” result. “Positive” is defined as greater than 10mIU/mL, in accordance with Centers for Disease Control and Prevention Guidelines and in agreement with the accepted level for protective immunity as defined by the World Health Organization.

  1. Was there any association with occult hepatitis B in those with or without anti-HBs?

We did not find an association with occult hepatitis B in our cohort. All transplant candidates are evaluated with complete hepatitis B serologic testing, including anti-HBs, hepatitis B surface antigen, and hepatitis B core antibody. If the results do not clearly correlate with the candidate being naïve to the virus or immune due to vaccination (i.e. an isolated positive hepatitis B core antibody), hepatitis B DNA is checked in order to rule out occult infection.

  1. Please provide detailed clinical demographics tables for your cohorts that includes ethnicity, ALT, AST, metabolic syndromes.

Additional information on metabolic comorbidities, including obesity, hypertension, diabetes mellitus, and hyperlipidemia, has been added, as requested. We did not add AST and ALT as these values are more variable over time. Each subject had AST and ALT measured on multiple occasions and the relevance of reporting a single measurement is unclear.

  1. Make sure to spell out the abbreviations when you first use them.

Abbreviations have been spelled out when first used in the Abstract, and again when first used throughout the body of the manuscript. We apologize for this oversight.

Reviewer 2 Report

The article “Room for Improvement: Vaccine Adherence in a Thoracic Organ Transplant Cohort” described by Jandhyala et al. is of great interest in discussing the usefulness of vaccination in thracic organ transplant medicine.

However, there are a number of things that need to be improved before it will be accepted for publication.

Major comments

  1. The title of the article“Room for Improvement: Vaccine Adherence in a Thoracic Organ Transplant Cohort” is not suitable because we cannot imagine the content of the paper easily. Because it's too vague, the authors need to give it a more specific title.

  1. Why did the authors choose HB vaccination and the pneumococcal vaccination in this study? There is no description about it.

  1. Only HB vaccine and pneumococcal vaccine are not sufficient, when discussing the usefulness of vaccines in thoracic organ transplant medicine. Please consider various vaccines in your study.

  1. This article does not mention the people who received the HB vaccine but were negative for anti-HBs. Please state how to handle them clearly. In discussion, please mention HB vaccine non-responder.

  1. How many individuals who have not received pneumococcal vaccines? How many individuals who are vaccinated with both PCV13 and PPSV23? There is no description about them. In that regard, Table 3 is very confusing.

  1. Table 3 is also hard to understand. 24/41 is not 70.6%. Please change to a table that is easy to understand. Also, this table has no meaning unless unvaccinated patients are added. There are also some mistakes in abbreviations.

Minor comments

  1. The expression "HBV vaccine" is not used. In many previous reports, a vaccine for preventing HBV infection is described as HB vaccine (HB vaccination).

  1. The meaning "HBV immunity" is not understood. Please correct it to an appropriate expression.

  1. In Line 57, the abbreviations PCV13 and PPSV23 have appeared. When the authors use the word for the first time in the article, please include the original word as well as the abbreviation.

  1. There are many miscellaneous mistakes are seen in this article. I will point out some examples. Pre-transplant TxID evaluation in Line 88, and HBV = hepatitis B vaccine at line 118, etc. Please fix them pretty carefully.

  1. In Materials and Methods, the definitions anti-HBs positive and negative are not listed. Please describe them.

  1. Regarding Table 1, the title is not suitable. Generally, describe it concretely like "Demographics of XX patients in this study". The expressions "Heart" "Lung" don't make sense for readers. Please make sure the formal disease names "Congenital" "Sarcoid".

Author Response

Thanks for your valuable comments

Major comments

  1. The title of the article “Room for Improvement: Vaccine Adherence in a Thoracic Organ Transplant Cohort” is not suitable because we cannot imagine the content of the paper easily. Because it's too vague, the authors need to give it a more specific title.

The title has been changed to “An analysis of adherence to vaccination recommendations in a thoracic organ transplant cohort”

  1. Why did the authors choose HB vaccination and the pneumococcal vaccination in this study? There is no description about it.

We intentionally included only inactivated vaccines in this report, such that vaccinations that were not completed pre-transplant but were completed post-transplant could be included in the analysis. The recombinant adjuvanted varicella zoster vaccine (Shingrix, GlaxoSmithKline Biologicals) was approved by the FDA in October 2017, which fell in the middle of our study period, so we did not feel it was appropriate to include in this analysis. Because the influenza vaccine is administered seasonally, rather than as a one-time dose, we felt its inclusion would not accurately reflect the impact of a one-time Transplant Infectious Diseases consultation. In addition, data analysis would be made quite complex by the seasonal availability of influenza vaccines, which would have to be factored into vaccination completion rates for subjects undergoing transplant infectious diseases evaluation or transplantation itself at varying times during the year.

  1. Only HB vaccine and pneumococcal vaccine are not sufficient, when discussing the usefulness of vaccines in thoracic organ transplant medicine. Please consider various vaccines in your study.

Please see response to #2 above for our rationale.

  1. This article does not mention the people who received the HB vaccine but were negative for anti-HBs. Please state how to handle them clearly. In discussion, please mention HB vaccine non-responder.

 We have updated the Results section to include results of anti-HBs testing post-vaccination. The following was also added to our Discussion, per request:

It is recommended that anti-HBs titers be checked approximately 4 weeks after the last dose of the hepatitis B vaccine, to ensure protective levels of anti-HBs have been achieved.[1] Just over half of our cohort had post-hepatitis B vaccination anti-HBs performed and we found significant rates of non-response, that is, a lack of development of protective immunity, as defined by an anti-HBs titer >10mIU/mL. In the heart transplant cohort, 1 of 5 subjects (20%) who had post-vaccination anti-HBs performed remained non-reactive, while in the lung transplant cohort, 9 of 20 subjects (45%) who had post-vaccination anti-HBs performed remained non-reactive. Guidelines recommend revaccinating hepatitis B vaccine non-responders with either a single booster dose of the vaccine or reinitiating the entire vaccine series.[1]

  1. How many individuals who have not received pneumococcal vaccines? How many individuals who are vaccinated with both PCV13 and PPSV23? There is no description about them. In that regard, Table 3 is very confusing.

We have updated and clarified Table 3, including subjects who received both pneumococcal vaccines. This is updated in the text as well (page 2, line 87 and page 3, line 98)

  1. Table 3 is also hard to understand. 24/41 is not 70.6%. Please change to a table that is easy to understand. Also, this table has no meaning unless unvaccinated patients are added. There are also some mistakes in abbreviations.

I believe the reviewer was referring to Table 4. Table 4 has been updated, including the denominator for each column, to make the percentages/rates more clear.

Minor comments

  1. The expression "HBV vaccine" is not used. In many previous reports, a vaccine for preventing HBV infection is described as HB vaccine (HB vaccination).

Throughout the manuscript, we have changed any use of “HBV” to hepatitis B, and thus the vaccine is described as “hepatitis B vaccine.”

  1. The meaning "HBV immunity" is not understood. Please correct it to an appropriate expression.

We have clarified in the Methods section that we are using a positive hepatitis B surface antibody (anti-HBs) to define hepatitis B immunity. Our clinical laboratory reports anti-HBs results as a qualitative “positive” or “non-reactive” result. “Positive” is defined as greater than 10mIU/mL, in accordance with Centers for Disease Control and Prevention Guidelines and in agreement with the accepted level for protective immunity as defined by the World Health Organization.

  1. In Line 57, the abbreviations PCV13 and PPSV23 have appeared. When the authors use the word for the first time in the article, please include the original word as well as the abbreviation.

This, as well as use of all abbreviations, has been carefully reviewed and corrected where necessary. We apologize for the oversight.

  1. There are many miscellaneous mistakes are seen in this article. I will point out some examples. Pre-transplant TxID evaluation in Line 88, and HBV = hepatitis B vaccine at line 118, etc. Please fix them pretty carefully.

We have reviewed the manuscript in detail to eliminate or correct any miscellaneous typos or mistakes. Again, we apologize for this oversight.

  1. In Materials and Methods, the definitions anti-HBs positive and negative are not listed. Please describe them.

In the Materials and Methods section, we have added how we have defined hepatitis B immunity, which is an anti-HBs titer greater than 10mIU/mL) , in accordance with Centers for Disease Control and Prevention Guidelines and in agreement with the accepted level for protective immunity as defined by the World Health Organization.

  1. Regarding Table 1, the title is not suitable. Generally, describe it concretely like "Demographics of XX patients in this study". The expressions "Heart" "Lung" don't make sense for readers. Please make sure the formal disease names "Congenital" "Sarcoid".

The title of Table 1 has been updated to: Demographics of Cohort of 96 Thoracic Organ Transplant Recipients. Disease names have been clarified as well.
